# Machine-Learning Functional Zonation Approach for Characterizing Terrestrial–Aquatic Interfaces: Application to Lake Erie

Léa Enguehard [1,2,*], Nicola Falco [1], Myriam Schmutz [3], Michelle E. Newcomer [1], Joshua Ladau [4], James B. Brown [4], Laura Bourgeau-Chavez [5] and Haruko M. Wainwright [1,6]

1   Earth and Environmental Sciences Area, Lawrence Berkeley National Laboratory, 1 Cyclotron Road, Berkeley, CA 94720, USA; nicolafalco@lbl.gov (N.F.); mnewcomer@lbl.gov (M.E.N.); hmwainwright@lbl.gov (H.M.W.)
2   ENSEGID, Bordeaux Institut National Polytechnique, 1 Allée F. Daguin, 33607 Pessac, France
3   EPOC UMR 5805, CNRS, Bordeaux INP, 1 Allée F. Daguin, 33607 Pessac, France; myriam.schmutz@bordeaux-inp.fr
4   Computational Biosciences Group, Lawrence Berkeley National Laboratory, 1 Cyclotron Road, Berkeley, CA 94720, USA; jladau@lbl.gov (J.L.); jbbrown@lbl.gov (J.B.B.)
5   Michigan Tech Research Institute, Michigan Technical University, Ann Arbor, MI 48105, USA; lchavez@mtu.edu
6   Department of Nuclear Science and Engineering, Massachusetts Institute of Technology, 77 Massachusetts Avenue, Cambridge, MA 02139, USA
*   Correspondence: leaenguehardlbnl@gmail.com

**Abstract:** Ecosystems at coastal terrestrial–aquatic interfaces play a significant role in global biogeochemical cycles. In this study, we aimed to characterize coastal wetlands with particular focus on the co-variability between plant dynamics, topography, soil, and other environmental factors. We proposed a functional zonation approach based on machine learning clustering to identify the spatial regions, i.e., zones that capture these co-varied properties. This approach was applied to publicly available datasets along Lake Erie, in the Great Lakes Region. We investigated the heterogeneity of coastal ecosystem structures as a function of along-shore distance and transverse distance, based on the spatial data layers, including topography, wetland vegetation cover, and the time series of Landsat's enhanced vegetation index (EVI) between 1990 and 2020. Results showed that the topographic metrics (elevation and slope), soil texture, and plant productivity influence the spatial distribution of wetland land-covers (emergent and phragmites). These results highlight a natural organization along the transverse axis, where the elevation and the EVI increase further away from the coastline. In addition, the clustering analysis allowed us to identify regions with distinct environmental characteristics, as well as the ones that are more sensitive to interannual lake-level variations.

**Keywords:** coastal wetlands; plant productivity; Great Lakes Region; machine learning; functional zonation; remote sensing

## 1. Introduction

Terrestrial–aquatic interfaces (TAIs) are dynamic transitions between land and water. Despite occupying small areas of the Earth's surface (0.07%–0.22%) [1], they play a significant role in global biogeochemistry and ecology [2]. TAIs provide countless ecosystem services, such as water purification, erosion retention, flood protection, and/or recreation [3]. TAIs are home to wetlands, which are highly productive ecosystems. However, TAI ecosystems are known to be one of the most vulnerable regions under climate change [4].

As one of the biggest freshwater TAIs, the Great Lakes Region (GLR), located between Canada and the United States, is the largest body of surface freshwater in the world (in surface area), and one of the most valuable natural resources on the Earth, holding nearly

20% of the Earth's unfrozen fresh surface water [5,6]. Coastal wetlands of the Great Lakes contribute to the health and maintenance of the GLR by playing major economic and ecological roles [7]. However, the GLR is experiencing substantial system changes and is heavily modified by anthropogenic activities [6]. Agriculture and other developments have already drained more than half of the wetlands in the Great Lakes Region, which makes the remaining wetland areas particularly vulnerable to climatic and human changes [8]. Plant species distribution among wetlands is influenced by water-level fluctuations. Recent extreme changes in the Great Lakes water levels (highest recorded lake levels in 2017 and 2019) are impacting coastal wetlands. Specifically, Smith et al. [9] showed that extreme high-water levels led to a decrease in vegetation coverage, and that low-water levels increase the coverage of invasive wetland species such as *Typha*.

There is a gap in our understanding of the spatial heterogeneity and key factors controlling wetland dynamics (such as topography, soil, plant productivity) on TAIs ecosystem [10]. The challenge is to characterize the coastal TAIs in an integrated and tractable manner, considering the strong spatial heterogeneity in geomorphology, soil, and vegetation communities. In addition, coastal ecosystems are characterized by sharp gradients in hydrology, soils, and vegetation. The difficulty to capture the spatial scales over which TAI gradients change could explain this knowledge gap. Although significant efforts to map wetland types in the GLR have been made, they are mainly carried out to manage and monitor these ecosystems in the TAIs [7]. Such integrated understanding is important to better incorporate the coastal systems in the Earth system models, as indicated by Ward et al. [10], who showed the coupled ecological and biogeochemical functions of TAIs.

In recent years, remote sensing and machine learning (ML) methods have been successfully applied to TAIs and equivalent systems. Several studies classified land cover and plant communities in coastal regions using classical ML methods, such as Random Forest (RF), Support Vector Machine (SVM), or Artificial Neural Network (ANN) [7,11,12]. In the GLR, Bourgeau et al. [7] mapped the different types of wetlands, including invasive species like *Typha* or *Phragmites.* However, most studies analyze spatial data layers for a single compartment such as a plant species map. These studies do not analyze how different layers (such as topography and plant productivity distribution) interact and co-vary each data layer. In addition, although there are studies that estimate soil biogeochemical properties based on remote sensing data [13], they have not been applied to the coastal region.

Unsupervised ML methods are considered powerful tools to improve the bedrock-to-canopy system understanding in the Earth and environmental sciences, which is often characterized by large unlabeled multidimensional datasets [14–16]. Clustering, in particular, reduces dimensionality in large datasets by extracting co-variability to identify zones of similar characteristics. This enables us to map the previously unknown structure of the dataset with a one-dimensional parameter that captures the dominant spatial heterogeneity [14,15]. Wainwright et al. [17] applied clustering to remote sensing and geophysical datasets of Arctic tundra ecosystems and found zones with specific ecosystem characteristics and terrestrial properties. Devadoss et al. [15] used unsupervised learning with time-lapse remote sensing images of plant productivity and identified zones with similar behaviors in terms of snow and plant dynamics. Wainwright et al. [18] used clustering to map the regions that have distinct watershed functioning, with watershed functions defined as drought sensitivity and nitrogen cycling. Using clustering again, Tu et al. [19] developed a framework including remote sensing imagery and mobile phone positioning data to identify urban functional zones. However, to our knowledge, this clustering zonation has not been applied to characterize coastal TAIs and particularly coastal wetlands of the GLR.

In this study, we proposed a coastal ecosystem functional zonation approach based on unsupervised ML to capture the self-organization and co-variability of the above/belowground terrestrial system, including soil texture, land cover, topographic metrics, and plant dynamics. In particular, we defined plant productivity of the coastal wetlands from satellite remote sensing data as an ecosystem function of interest. We follow recent studies

to characterize the spatial variability of plant dynamics based on long-term historical satellite images [20,21]. In addition, we explored their dependence on the along-shore distance and transverse distance, since these distances are drastically modified by the geomorphic features of the estuaries and lakes on which they are superimposed. We demonstrated our approach with publicly available datasets from Lake Erie, which is the southernmost Great Lake. In particular, we take advantage of the wetland types that were defined and delineated recently based on optical satellite images [7].

We hypothesize that (1) the different wetland types have distinct topographic signatures and plant productivity, and are organized as a function of the distance from the shore; and (2) we can find statistically distinct zones that capture these co-varied properties among the coastal wetlands. To test this hypothesis, we used two methodologies. First, we investigated the co-variability between topographic metrics, plant productivity, and soil data across the coastal wetlands in order to characterize them. Second, we applied a specific unsupervised ML method, called hierarchical clustering, to establish spatial zones of similar topography and plant productivity characteristics. Testing this hypothesis is crucial to bringing understanding of the spatial heterogeneity of wetland on TAIs, and therefore to help monitor and manage these ecosystems. It is also important to show that unsupervised learning is an efficient way to synthesize multiple spatial data layers, and to create functional spatial zones.

## 2. Materials and Methods

### 2.1. Description of the Study Site

Lake Erie is the fourth-largest lake of the five Great Lakes (Figure 1). Lake Erie is located at the border between two countries, with its north coast in Canada and south coast in the United States. The major axis of the lake extends from west–southwest to east–northeast for 392 km, and its minor axis extends from north–northwest to south–southeast for 90 km at its most. Lake Erie's shoreline is 1402 km long, and home to major natural ecosystems and U.S. cities. Buffalo (New York) and Toledo (Ohio) are located at the tip of each side of the lake. Cleveland (Ohio) is located on the south coast, and Detroit (Michigan) is at the junction of Lake St. Clair and Lake Erie.

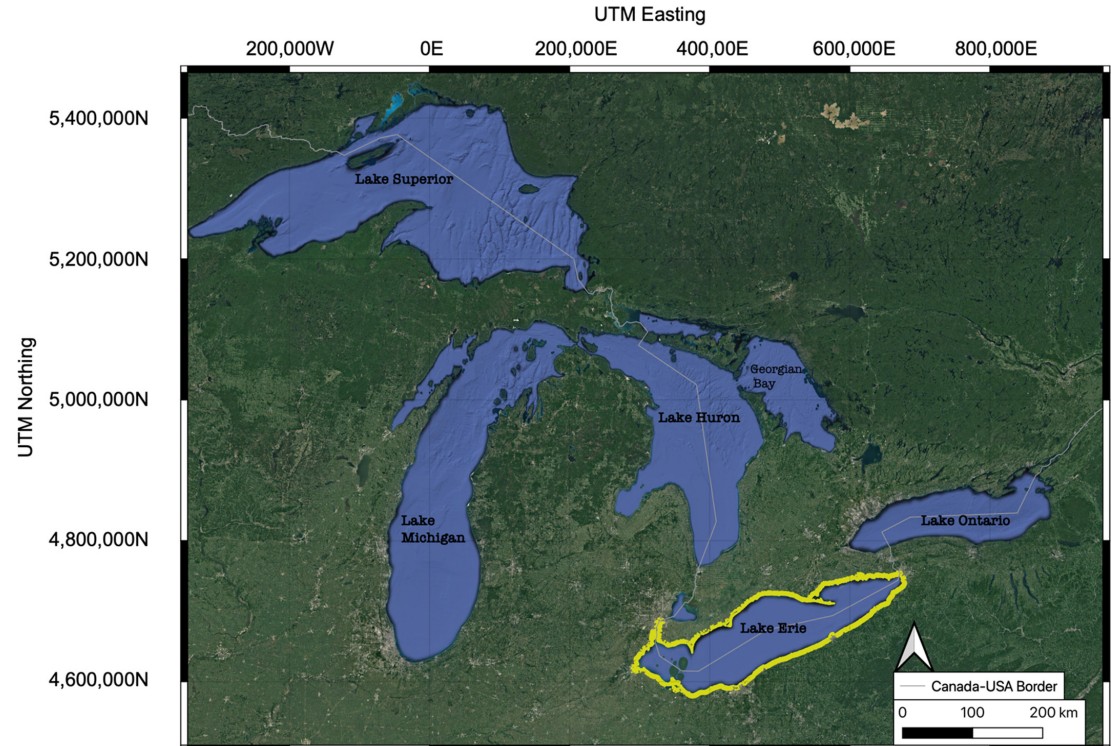

**Figure 1.** The Great Lakes Region (satellite image from Google Maps). Lake Erie is highlighted with a yellow band.

Lake Erie's primary water source comes from Lake Huron, the upper Great Lake, via the Detroit River (90%) [22]. Several watersheds drain into the lake as well, primarily located in the western region of the lake [22]. Outflow is through the Niagara River in Buffalo, and the Welland Canal diversion [22]. Lake Erie's average water level is about 174 m above mean sea level, fluctuating between seasons (high in June, low in February), but drastically increasing over the past decade [22,23]. With a maximum depth of 64 m, and an average depth of 19 m, Lake Erie is the shallowest lake of all the Great Lakes, and one of the most biologically productive [22]. Lake Erie has warm summers and cold winters (with an average air temperature of 23.5 °C in summer and −3 °C in winter in Erie, PA) marked by severe storms, snowfall, and rapid changes in water level [22].

### 2.2. Spatial Data Layers

#### 2.2.1. Land Cover Map

In our study, we used the land-cover map of Lake Erie computed in 2014 using satellite images (LANDSAT and PALSAR) and extensive fieldwork by Michigan Technical University (MTU) [7]. The mapped area stretches from the coastline to 10 km inland, to capture all the TAIs at a spatial resolution of 20–30 m, with an overall map accuracy of 92% [7]. The map represents 23 categories of land covers, with an emphasis on wetland types. We analyzed the land-cover map and computed the surface area of each different land type. We particularly focused on wetland ecosystems, which are divided into four distinct land-cover classes: forested wetland, shrub wetland, emergent wetland, and phragmites [7]. Because the land-cover map was made using only satellite imagery, our analysis with topographic metrics is independent.

In the MTU classification, the forested wetlands class is defined as "wetlands dominated by woody vegetation (dead or alive) superior to 6 m in height, including seasonally flooded forests. The crown closure percentage (i.e., aerial view) is more than 50%". Shrub wetlands are defined as "wetlands dominated by shrubs inferior to 6 m in height. The crown closure percentage is superior to 50%". Finally, the emergent wetlands class is de-

fined as "emergent wetland and wet meadow vegetation not represented by other classes, and seasonal inundation and/or drying are common phenomena". In addition, we considered the class "phragmites" as an invasive wetland monoculture usually associated with emergent wetlands.

### 2.2.2. Topographic Metrics

We used the Shuttle Radar Topography Mission (SRTM) Digital Elevation Model (DEM) to derive topographical properties. The SRTM DEM was produced on 11–22 February 2000, aboard the Endeavor space shuttle using interferometric radar. The shuttle orbited Earth 16 times during the mission and collected over 80% of the Earth's land-surface radar data (between 60°N and 56°S latitude) with a spatial resolution of 30 m. In addition to the elevation, we computed the slope from the DEM based on the gradient function in MATLAB (Natick, MA, USA).

### 2.2.3. Time-Series Landsat Satellite Images

To study plant dynamics on Lake Erie TAIs, we used satellite images acquired from Landsat (NASA) satellites. Landsat provides data at about 16-day intervals and at a spatial resolution of 30 m/pixel. Vegetation indices derived from atmospherically corrected surface reflectance in the red, near-infrared, and blue wavebands reveal particular characteristics of vegetation. They are robust, empirical measures of vegetation activity at the land surface [24]. Two VIs are commonly used to monitor vegetation at global and local scales in all ecosystems and climates: Normalized Difference Vegetation Index (NDVI) and Enhanced Vegetation Index (EVI). The latter is an ameliorated version of the NDVI, which has improved sensitivity to high biomass regions and improved vegetation monitoring [24,25]. EVI values range from −1 to +1, with healthy vegetation varying between 0.2 and 0.8. In our study, we used Google Earth Engine to gather a time-series corresponding to 31 years (1990–2020) and computed for each year the annual maximum EVI to capture the yearly peak plant productivity which is the main functional attribute of interest.

### 2.2.4. Soil Texture

We used soil data from the National Cooperative Soil Survey (NCSS), operated by the United States Department of Agriculture (USDA). NCSS provides a map of the soils found in the Lake Erie region, with detailed characteristics of each soil type, such as the taxonomic class, typical pedon, color, soil temperature, and geographic settings. Due to the data being collected by U.S. organizations, there is no soil information on the Canadian side of Lake Erie.

To simplify this highly compressed map, we classified the soils based on their texture. Soil texture is determined by the percentage of sand, silt, and clay within the soil—sand particles being the coarsest and clay the finest. We established our classification with the soil characteristics provided by NCSS and a texture triangle. We simplified the soil-texture classification and considered that the soil is silty, sandy, or clayey when it contained more than 50% respectively of silt, sand, or clay (Figure S1). All four data layers, including the EVI layer discussed here, are shown in Figure 2.

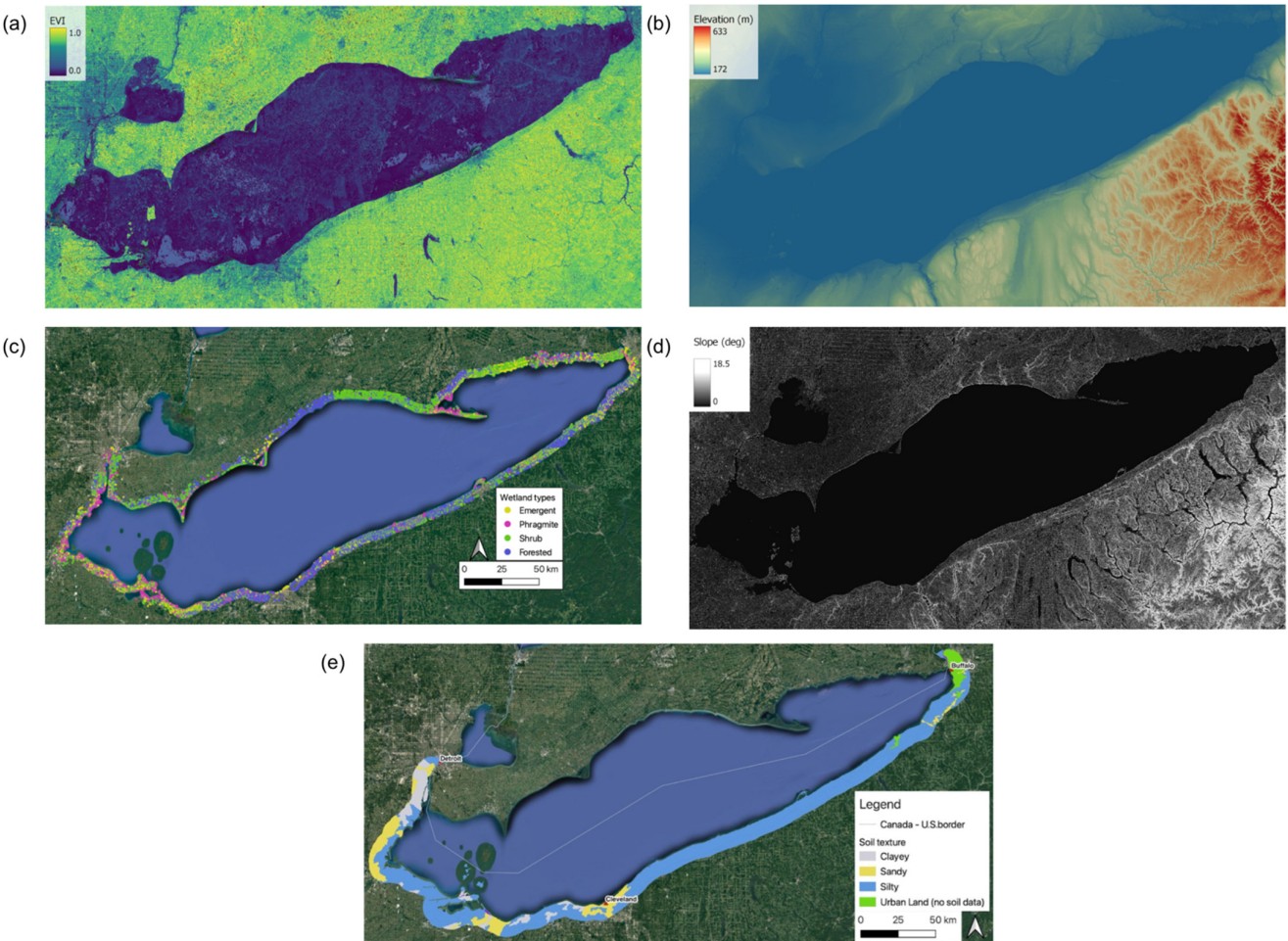

**Figure 2.** Spatial data layers used for the study: (**a**) EVI (used as proxy for plant productivity); (**b**) elevation; (**c**) wetland land cover; (**d**) slope; (**e**) soil texture.

### 2.3. Zonation Analysis

#### 2.3.1. Grid

We resampled the various data layers into a common two-dimensional grid to better investigate the variability of the data layers across both longitudinal and transverse dimensions (Figure 3a). We developed a MATLAB algorithm to build a grid based on principles of geometry, made of lines perpendicular and parallel to the shoreline (i.e., transverse and longitudinal dimensions). The grid points (intersections between the transverse and longitudinal dimensions) are 100 m apart on the shoreline of Lake Erie, and the grid extends 5 km inland and along the entire coast of the lake.

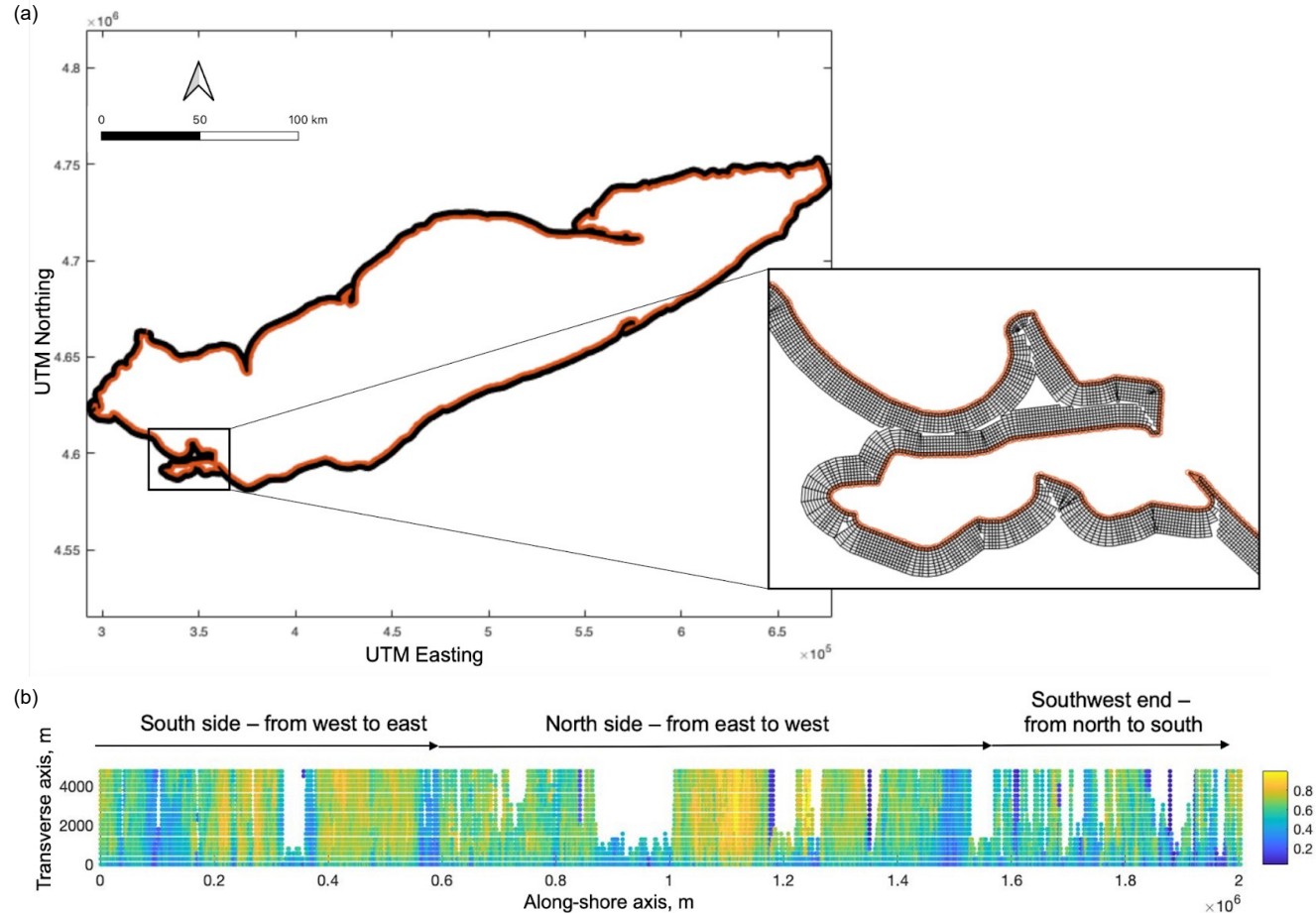

**Figure 3.** (**a**) Lake Erie's grid, the entire grid on the left, and zoom on the Portage River region, where we can see the transverse and longitudinal axis on the right; (**b**) scatter plot of the 2001 peak EVI extracted on the grid. The *x*-axis is the distance along the shoreline, and the *y*-axis is the distance from the shoreline. We can see some regions without points, which are located in peninsulas.

We then extracted all the data layers at each grid point. Thus, each point of intersection between the longitudinal and transverse axis contains a value of land cover (categorical data), topographic metrics (elevation and slope), EVI, and soil type (categorical data; only on the U.S. side of the lake). We carried out all statistical analysis in this grid as a function of distance from the shore (transverse axis), and distance along the shore (longitudinal axis) (Figure 3b). When we analyzed wetlands properties, we only kept grid points associated with the wetland's land cover (phragmites, emergent, shrub, and forested wetlands).

There are some regions without any values on the grid, leaving blanks on the scatter plot, which are computing errors due to the shape of the coastline (narrow and windy coastline, which makes the grid superimposed on both sides). We assumed these errors were not significant for this study, because there were still many diverse points with which to perform our analysis.

### 2.3.2. Unsupervised Clustering for Functional Zonation

We applied agglomerative hierarchical clustering, an unsupervised ML method, to group the TAI features (topographic metrics, EVI, soil type, etc.) and identify zones with similar parameters on Lake Erie TAIs (Figure 4). Agglomerative hierarchical clustering is a tree-based method that uses a measure of distance between pairs of observations to group similar data [15,26]. Because it is agglomerative, the algorithm follows a bottom-up strategy. It starts by considering each data point as an individual cluster (zone), and

step-by-step merges clusters at an increasing level of similarity until data points form one large cluster [26]. This tree-like path can be observed through a dendrogram computed by the algorithm.

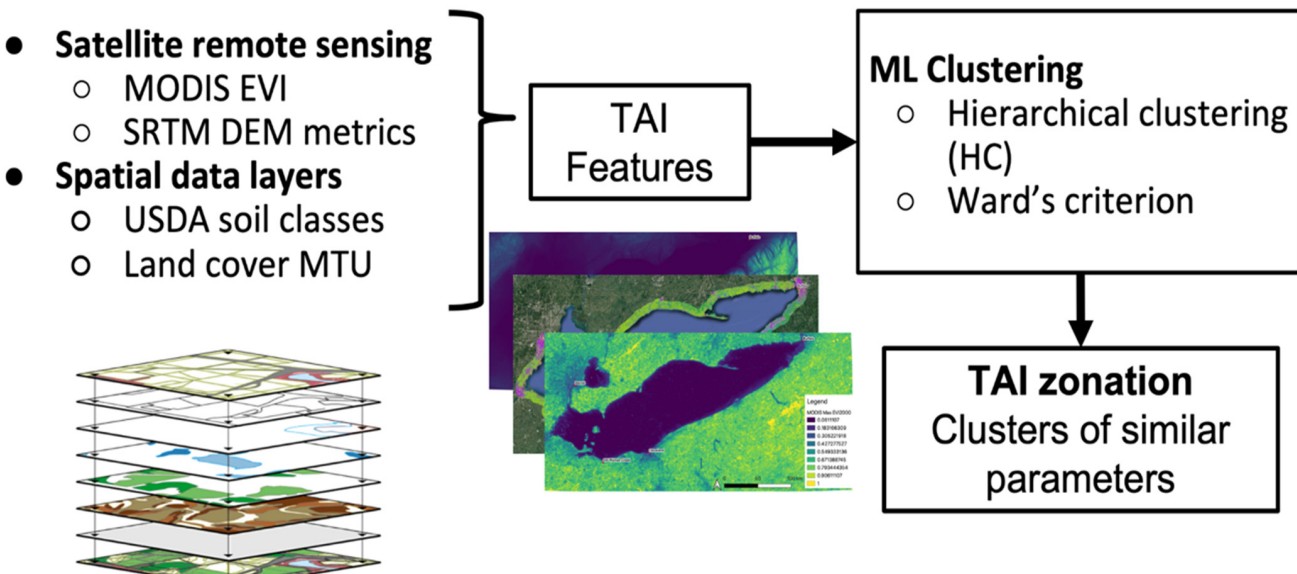

**Figure 4.** ML hierarchical clustering process.

There are several distance criteria with which to apply hierarchical clustering; we chose Ward's criterion [27], which has been successfully used for Earth systems in the past [15]. It consists of minimizing the total variance within each cluster. Ward's algorithm finds a pair of clusters at each step, leading to the minimum variance within each cluster after merging. We determined the number of clusters from the highest level of clustering, which shows the most dissimilarities between clusters.

We performed two different zonation analyses: the first zonation (all-metrics-based zonation) integrates the maximum EVI time-series from 1990 to 2020, with elevation and slope to investigate how plant productivity and topographical metrics co-vary along the TAIs; the second zonation (EVI-based zonation) includes the maximum EVI time-series from 1990 to 2020 only and aims to investigate the spatial variability of plant productivity which is the TAI function of interest.

## 3. Results

### 3.1. Lake Erie's TAI Land Cover and Soil Texture

Agricultural land, forests, and suburban land (primary residential areas where man-made structures are present with more than 25% vegetation interspersed [7]) contribute to the majority of Lake Erie's TAIs, with respectively 33.3%, 11.4%, and 12.8% surface-area coverage. Urban lands (urban, urban grass, urban road) represent approximately 10% of the TAIs, while other naturally vegetated lands (wetlands, orchards, fallow fields, pine plantations, etc.) cover about 24% of the surface. Agricultural fields are mainly located on the north shore and west side of Lake Erie. Forests are for the most part situated on the south shore, between Cleveland and Buffalo, but they are also sparsely present on the northeast and southwest sides of the lake. Urban and suburban lands are concentrated in the major cities and their suburbs: Detroit, Buffalo, Cleveland, and Toledo, with urban roads connecting them.

Wetlands (phragmites, emergent, shrub, and forested wetlands) cover approximately 13% of Lake Erie's TAIs. Among the four wetland types, forested wetlands represent almost half of the TAI (40%), shrub wetlands another significant part (39%), and phragmites and emergent wetlands the smallest part (respectively 9% and 11%). Forested wetlands are

mainly located on the south shore, next to the forest class. Shrub wetlands are evenly distributed around the lake, but are nonexistent next to urban and agricultural regions, specifically on the north shore. Emergent wetlands are concentrated along the western part of the lake, where the lake is shallower, and at the eastern end, close to the lake drainage points. Phragmites are usually associated with emergent wetlands.

Using the NCSS soil data, we grouped the soils into three categories based on their texture: silty, clayey, and sandy. On the south coast of Lake Erie, silty soils are the most abundant soil texture. Sandy and clayey soils are predominantly located along the western end of the lake.

*3.2. Wetland Characterization*

We characterized Lake Erie's coastal wetlands as a function of distance from the shoreline, topographic metrics, and plant productivity. This analysis helped obtain further insight into wetlands heterogeneity across the TAIs.

We first investigated the distance from the shoreline and associated wetland types. We extracted the grid points associated with wetlands, computed their distance to the shoreline, and distinguished the four types of wetlands (phragmites, emergent, shrub, and forested wetlands) (Figure 5a). The results showed that phragmites and emergent wetlands are located closer to the shoreline than any other wetland type, with a median of 750 and 1000 m. Shrub wetlands are further away from the lake (median of 1300 m), and forested wetlands are the furthest away from the shoreline (median of 1800 m).

Secondly, we examined the elevation and associated wetland types. We extracted the Digital Elevation Model (DEM) from the grid and defined the elevation of each wetland type (Figure 5b). The southeast side of Lake Erie is hilly and has the highest elevation around the lake. Phragmites and emergent wetlands are the closest to the lake level (median at 176 &183 m), shrub wetlands are slightly higher in elevation than emergent wetlands (median at 185 m), and forested wetlands are the highest (median at 200 m). There are a significant number of outliers with higher elevations, which corresponds to the hilly part of the lake.

To continue, we explored the slope degree and associated wetland types. We computed the slope from the DEM and extracted it from the grid. We then plotted the slope as a function of wetland types (Figure 5c). The entire TAI does not show a lot of variation in terms of slope and is relatively flat, varying from $0°$, close to the shoreline, to $14°$, in the hilly region. Emergent wetlands and phragmites are located on flat regions ($0°$ to $0.6°$), whereas shrub wetlands and forested wetlands are on steeper slopes but still relatively flat ($0°$ to $2°$). Forested wetlands show many outliers with steeper slopes, corresponding to the hilly part of the lake.

Finally, we considered the plant productivity of each wetland type. We extracted the maximum annual value of EVI at each point of the grid for the 31 years (1990–2020) time-series dataset. To define a range of values for each of the three wetland types, we computed the average annual peak EVI over the 31 years for each wetland point (Figure 5d). The results showed that forested wetlands have the highest values of EVI, with a median of 0.65, and that emergent wetlands and phragmites have the lowest values of EVI, with a median of respectively 0.54 and 0.50. Shrub wetlands EVI values are in between emergent and forested wetlands, with a median of 0.60. This shows that forested wetlands are more productive ecosystems than shrub and emergent wetlands, and that the latter is the least productive one. The characteristics of each wetland type is shown in Table 1. The ANOVA test has confirmed that the wetland types have significantly different mean values for these metrics ($p$-values $< 1 \times 10^{-10}$).

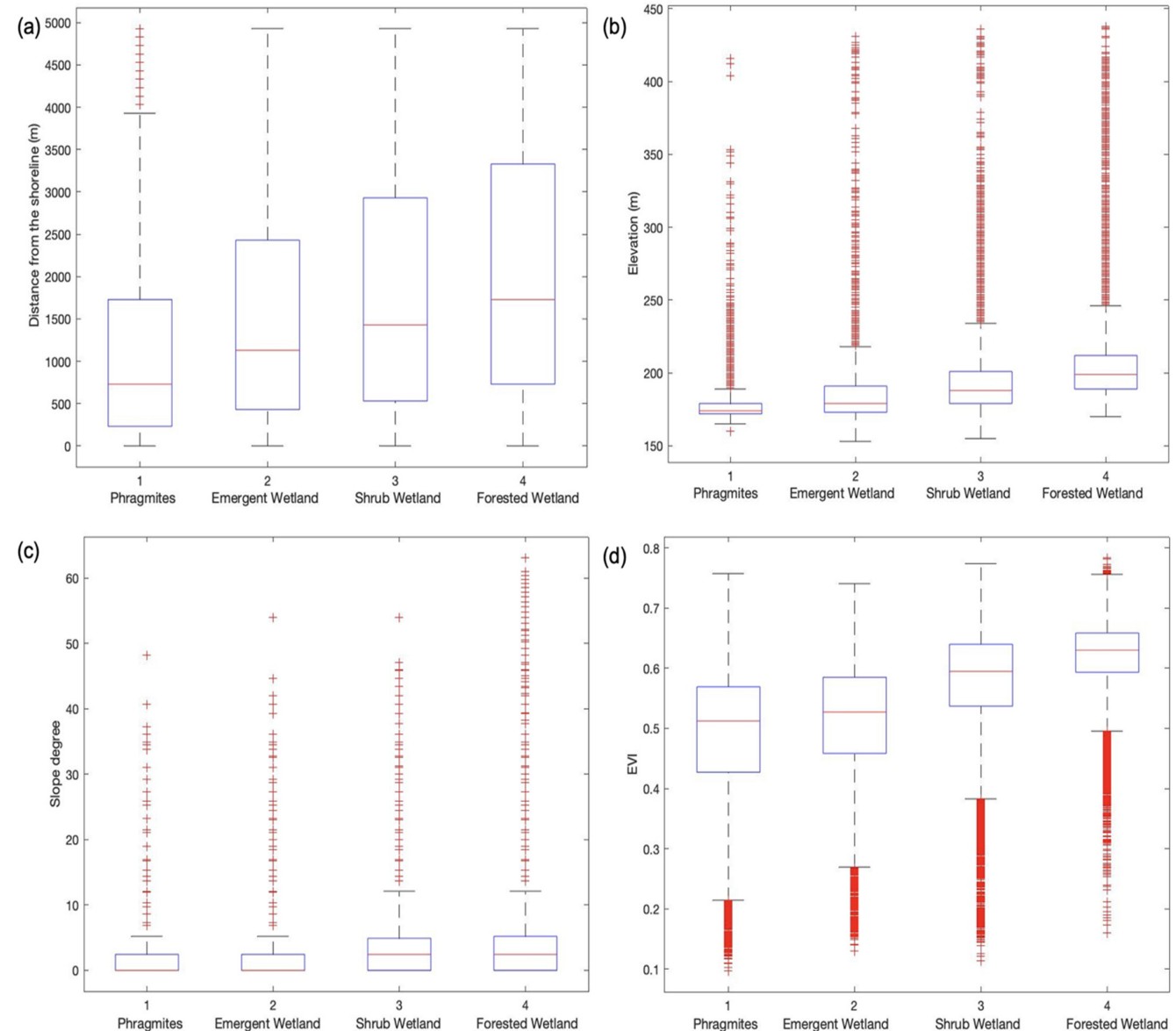

**Figure 5.** Wetland types as a function of: (**a**) distance from the shore; (**b**) elevation; (**c**) slope degree; (**d**) EVI.

**Table 1.** Characteristics of each wetland type, where each value corresponds to the median of the group.

|  | Phragmite | Emergent Wetland | Shrub Wetland | Forested Wetland |
|---|---|---|---|---|
| **Distance from the shore** | 750 m | 1000 m | 1300 m | 1800 m |
| **Elevation** | 176 m | 183 m | 185 m | 200 m |
| **Slope** | 0° | 0.6° | 1.0° | 1.5° |
| **Enhanced Vegetation Index** | 0.5 | 0.54 | 0.60 | 0.65 |

### 3.3. Functional Zonations

For both all-metrics-based and EVI-based zonations (Figure 6), we selected the grid points associated with phragmites and emergent wetlands and applied clustering with these points only. In this paper, we specifically focused on phragmites and emergent wetlands, as they are significantly closer to the coast than forested and shrub wetlands,

and our goal was to study the coastal zone. In addition, given that emergent wetlands and phragmites are closer to the shore, they are more sensitive to the lake's water level. Hence, focusing on these wetlands is useful for identifying regions that are most sensitive to climate change.

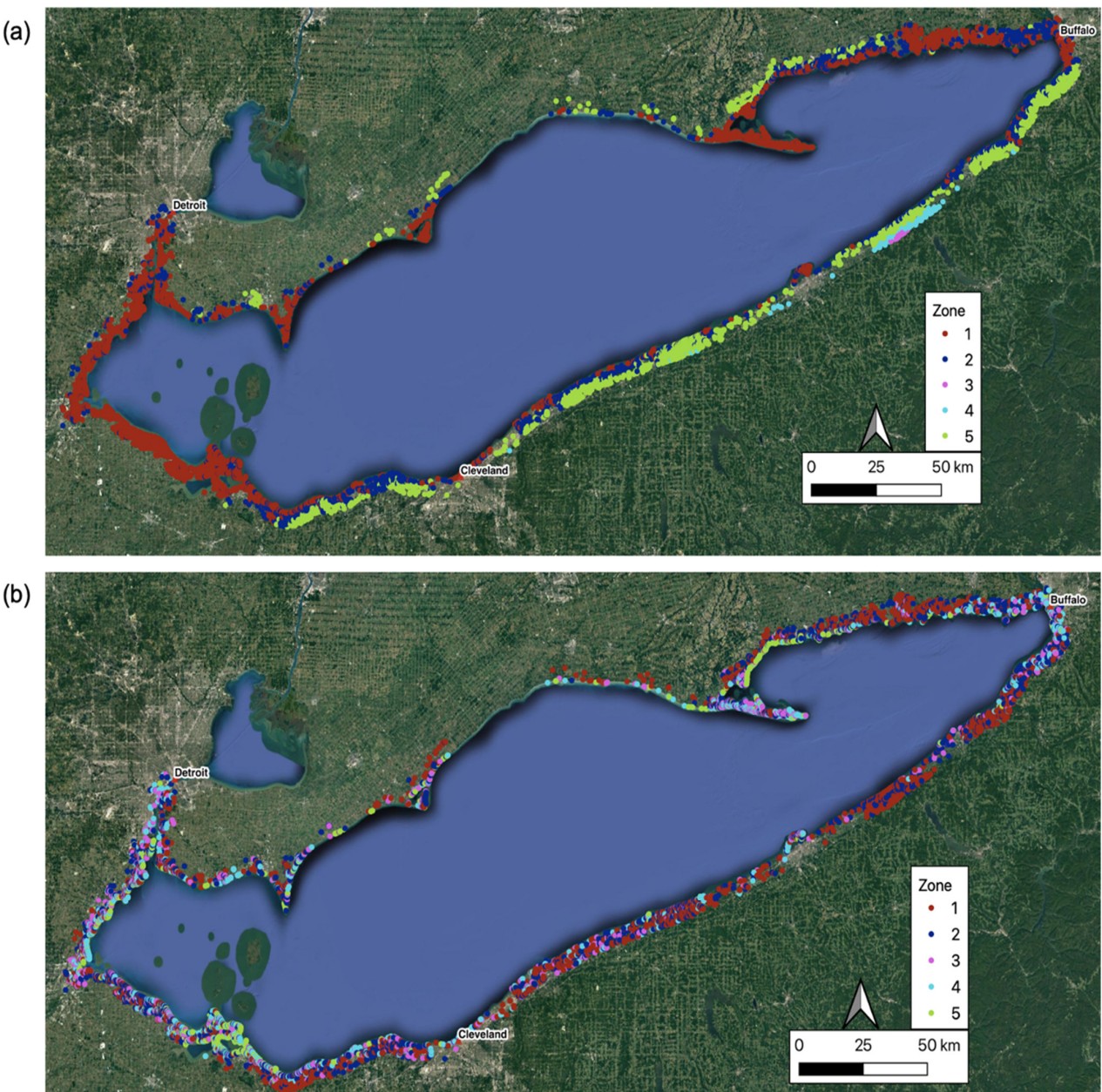

**Figure 6.** Results of the functional zonations: (**a**) map of Lake Erie's all-metrics-based zonation; (**b**) map of Lake Erie's EVI-based zonation.

### 3.3.1. All-Metrics Based Zonation

There were five zones identified by clustering applied to all the metrics (Figure 6a). Zone 1 is the low-lying western region, as well as peninsulas. Zone 2 is the transition between the coast and inland. Zones 3, 4, and 5 are limited to inland areas.

We analyzed the topographic metrics within the zones (Figure 7a,b). We found that each zone could be defined by a specific value of elevation and slope. In terms of elevation, Zone 3 was the highest one (median 408 m), Zone 4 was second to highest (median 284 m), Zone 5 was lower than Zone 4 (median 208 m), Zone 2 was second to lowest (median

186 m), and Zone 1 had the lowest elevation, close to lake level (median 173 m). The slope followed roughly the same order as the elevation, with elevated areas steeper than low-lying lands.

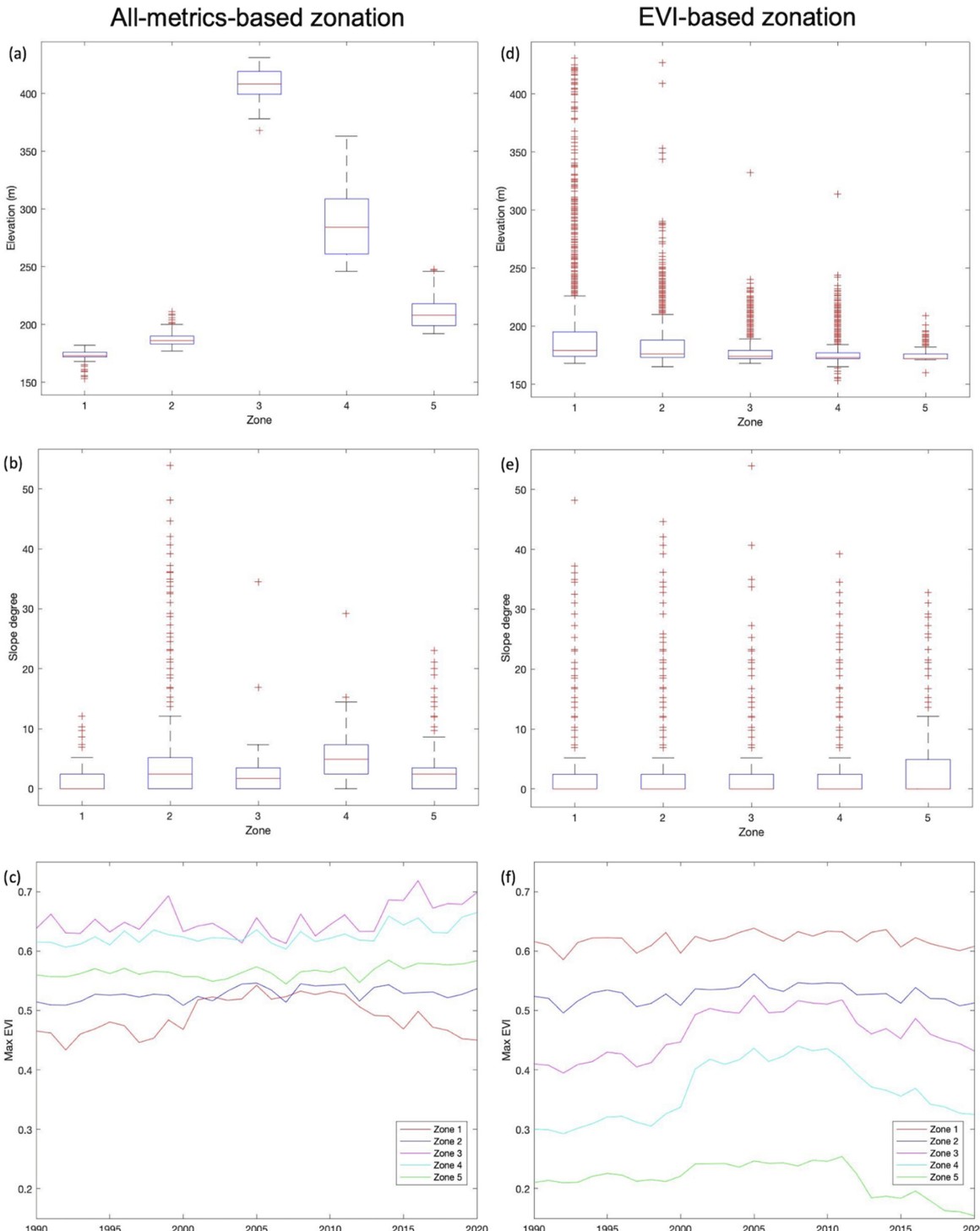

**Figure 7.** Distribution of the variables within the clusters of two zonations: the left column is all-metrics-based zonation, the right column is EVI-based zonation. The variables are: (**a**,**d**) the elevation; (**b**,**e**) the slope; (**c**,**f**) yearly mean EVI for the period 1990–2020. Elevation, slope, and EVI are all statistically different in each zone.

Secondly, we computed the yearly EVI mean for each zone. We found that they were defined by a specific trend and average value of EVI (Figure 7c). Zone 3 had the highest EVI (mean 0.65), followed by Zone 4 (mean 0.63). Zones 2 and 5 were in the middle (mean 0.56 and 0.53), and Zone 1 was characterized by the lowest trend and smallest values (mean 0.49). Zones 1 and 2 showed increasing EVI values between 2000 and 2015.

Finally, we added each pixel of soil type for each zone in a bar plot (on the US side of Lake Erie) (Figure S2). All the zones were mainly silty, but Zone 1 had more clay and sand than the other zones (Table 2).

**Table 2.** Characteristics of each zone from the all-metrics-based zonation. Slope and EVI values correspond to the mean of the group, and elevation values to the median. The ANOVA test confirmed that these five zones are significantly different for these metrics ($p$-values $< 1 \times 10^{-10}$).

|  | Zone 1 | Zone 2 | Zone 3 | Zone 4 | Zone 5 |
|---|---|---|---|---|---|
| Location | Low-lying lands, peninsulas | Transition between inland and the water | Inland, high hilly region | Inland, hilly region | Inland |
| Predominant soil type | Silty, sandy, clayey | Silty, sandy, clayey | Silty | Silty | Silty, clayey |
| Elevation | 173 m | 186 m | 408 m | 284 m | 208 m |
| Slope | 1.1° | 4.6° | 3.1° | 5.4° | 2.3° |
| Peak EVI | 0.49 | 0.53 | 0.65 | 0.63 | 0.56 |

### 3.3.2. EVI-Based Zonation

The EVI-based zonation shows zones that are more scattered than with the all-metrics based zonation (Figure 6b).

We analyzed the average EVI for each zone (Figure 7f). We found two higher plant-productivity regions contained in Zones 1 and 2 (mean of 0.62 and 0.53), and two lower plant productivity regions contained in Zones 3 and 4 (mean of 0.46 and 0.36). Zones 3 and 4 showed a particularly strong EVI increase between 2000 and 2015. We also identified Zone 5 with abnormally low values, which we associate with landcover misclassifications, where the grid points are often located on water or on human infrastructure (e.g., bridge, building). We noticed that low plant-productivity regions were located closer to the shoreline than high plant-productivity regions. We also investigated how the topographic metrics are spread across the clusters, even though they were not input layers of this EVI-based zonation (Figure 7d,e). We found that high plant-productivity regions were at a higher elevation than lower plant-productivity regions (Table 3).

**Table 3.** Characteristics of each zone from the EVI-based zonation. Slope and EVI values correspond to the mean of the group, and elevation values to the median. The ANOVA test confirmed that these five zones are significantly different for these metrics ($p$-values $< 1 \times 10^{-10}$).

|  | Zone 1 | Zone 2 | Zone 3 | Zone 4 | Zone 5 |
|---|---|---|---|---|---|
| Description | Highest plant productivity regions | High plant productivity regions | Medium plant productivity regions | Low plant productivity regions | Water/misclassification |
| Predominant soil type | Silty, clayey | Silty, clayey, sandy | Silty, clayey, sandy | Silty, clayey, sandy | Silty, sandy |
| Elevation | 179 m | 176 m | 174 m | 173 m | 172 m |
| Slope | 2.0° | 1.9° | 1.5° | 1.8° | 3.2° |
| Peak EVI | 0.62 | 0.53 | 0.46 | 0.36 | 0.22 |

## 4. Discussion

Results showed that the wetland types were organized along the transverse direction, influenced by the co-varied nature of the topography and the plant types. Topographic metrics (elevation and slope) influence the spatial distribution and connectivity of wetland land cover (phragmites, emergent, shrub, and forested wetlands), as well as plant productivity (i.e., EVI as a proxy). Emergent wetlands and phragmites were close to Lake Erie's shore, in low-lying lands, and with low plant productivity. Shrub wetlands were further away from the shore, higher in elevation compared with emergent wetlands, and with higher plant productivity. Forested wetlands were the furthest away from the lake,

the highest in elevation, and with much higher plant productivity than the other wetland types. These results highlight the spatial organization of vegetation along the transverse axis, where, generally, the elevation and the EVI increase further away from the coastline (Figure 8).

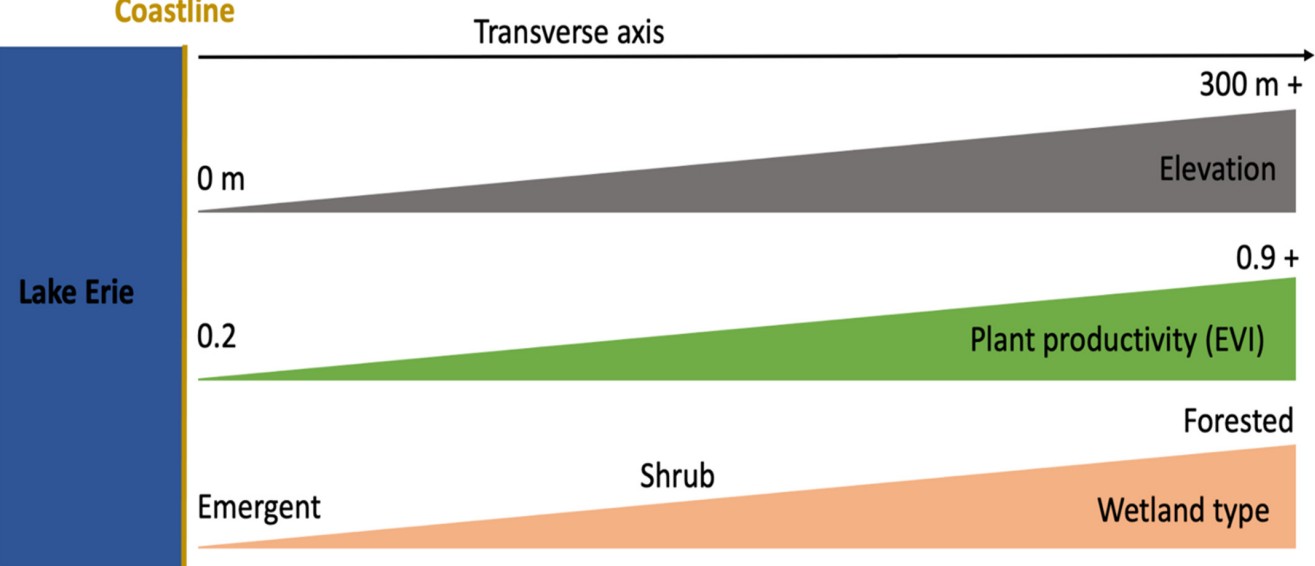

**Figure 8.** The coastal structure along the transverse axis.

We observed that the plant productivity, i.e., EVI, increased in some zones (Zones 1 and 2 in Figure 6a; Zones 3 and 4 in Figure 6b) from 2000 to 2015 (Figure 7c,f). Although have analyzed potential climate factors, such as annual total precipitation and annual temperature (min, max, mean), we did not find any correlation between the EVI and these factors. The increase from 2000 to 2015, however, corresponds to the low lake level (Figure 9a). The increase was more apparent in the regions that are close to the lake (Zone 1 in Figure 6a, and Zone 4 in Figure 6b). This represents the impact of the lake water level on the plant dynamics of the TAIs.

The zonation analysis allowed us to identify coastal zones with specific environmental characteristics, particularly within the low-lying wetland types with similar features (including emergent wetlands and phragmites). The zones capture the heterogeneity of co-varied plant productivity and topographic properties, such that the high-elevation zones could be associated with high plant productivity.

We considered two different zonations: the all-metrics-based, including time-series plant productivity and topographic metrics, and the EVI-based, including time-series plant productivity only (Figure 6). We observed that the all-metrics-based zonation yields spatial zones with a large-scale structure characterized by extensive homogeneous regions. On the other hand, the EVI-based zonation identifies different zones in close proximity. This is because plant productivity has small-scale variability affected by different land types and species types, compared with topography and other metrics. However, both zonations provide convergent results, with low plant-productivity areas located at lower elevations than high plant-productivity areas. Because the all-metrics-based zonation includes plant productivity and topography, it appears to give a better understanding of Lake Erie's TAI organization at a large scale, while the EVI-based zonation emphasizes plant-productivity patterns that have higher variability at a local scale.

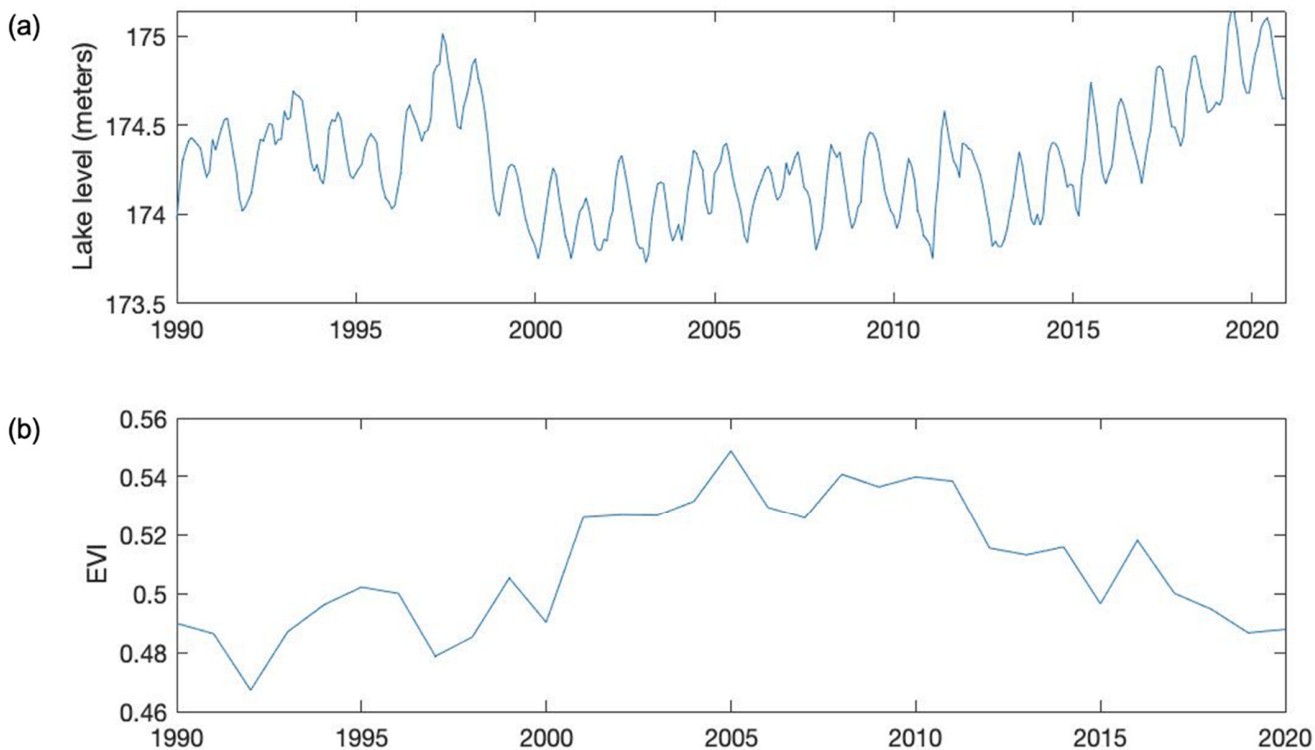

**Figure 9.** (**a**) Monthly mean lakeside average water levels from Toledo and Cleveland gauges (NOAA); (**b**) average max EVI for all phragmites and emergent wetlands of Lake Erie's TAI.

Looking at the EVI temporal characteristics, both zonations identify zones where EVI increased in the period 2000–2015 and then decreased afterward (Zone 1 in all-metrics-based zonation and Zones 3 and 4 in EVI-based zonation). These zones are associated with areas close to the shore that are susceptible to lake level. The EVI temporal variation indeed seems to follow the lake water-level dynamic (Figure 9a,b) with EVI increasing with the decrease in water level. However, this effect seems to gradually diminish in the zones as the distance increases from the shoreline and higher elevation.

Our approach differs from those used in other ML studies of the Great Lakes region. Bourgeau et al. [7] focused on plant species, while we focused on how different spatial layers, such as topography and plant-productivity distribution, interact and co-vary together. Moreover, while the hierarchical clustering method we applied in this study has been used in the past [15,17], our zonation brings additional insights into plant dynamics and its sensitivity to the lake-level variations, identifying the high-sensitivity regions even within the same wetland class.

## 5. Conclusions

This study investigated the spatial heterogeneity of coastal TAIs and used a clustering method to establish spatial zones of similar topographic characteristics and plant productivity. We found that there is a natural organization along the transverse axis, where the elevation and the wetlands EVI increase further away from the coastline. The clustering analysis allowed us to identify regions with distinct environmental characteristics. In particular, our analysis identified the regions where wetlands plant productivity is more susceptible to lake-level variations, hence more vulnerable to climate change. As the lake water level varies, sediment and nutrient deposition rates will shift accordingly, which has implications for wetland vegetation establishment and nutrient cycling [28]. The use of a grid helped us investigate these heterogeneities along the transverse and longitudinal gradients.

We acknowledge that our work has some limitations. The land-cover map has an accuracy of 92%, but we found misclassification in some regions [7]. For instance, some wetlands and other vegetated land covers are located on human infrastructure, such as bridges or roads, and on the lake water. Additionally, the land types were classified in 2014, and in our study, we used the same map for the entire period from 1990 to 2020. There is the possibility that some land types have changed during that time. Furthermore, in terms of spatial resolution, our analysis grid is 100 m in resolution, and the DEM and Landsat satellite images (EVI) is 30 m, so some small-scale variability may not have been captured.

Nonetheless, an unsupervised machine-learning-based coastal functional zonation approach is still a powerful way to synthesize multiple available spatial data layers, and to establish spatial zones of similar topographic characteristics and similar plant dynamics. Our approach is general, applicable to other sites, and extendable. There is the potential to add additional data layers or to employ higher resolutions as more datasets are available, which can refine zonations and our understanding of the spatial structure of TAIs. Our research can also be important for informing site selections and in performing representative analysis before collecting any site-specific data, which would be particularly useful to organizations and local communities that manage and monitor these ecosystems.

**Supplementary Materials:** The following supporting information can be downloaded at: https://www.mdpi.com/article/10.3390/rs14143285/s1, Figure S1: Soil texture triangle (USDA soil texture triangle); Figure S2: Percentage of soil texture in each cluster for: (a) the all-metrics based zonation; (b) the EVI-based zonation.

**Author Contributions:** Conceptualization, H.M.W., L.E., N.F., M.S.; methodology, H.M.W., L.E., N.F.; software, L.E., H.M.W., N.F.; validation, L.E., N.F., H.M.W.; formal analysis, L.E.; investigation, L.E., N.F., H.M.W.; resources, H.M.W., N.F., L.B.-C.; data curation, N.F., L.E., H.M.W., L.B.-C.; writing—original draft preparation, L.E.; writing—review and editing, L.E., H.M.W., N.F., M.S., J.L., M.E.N.; visualization, L.E., N.F.; supervision, H.M.W.; project administration, H.M.W., N.F.; funding acquisition, H.M.W., J.B.B. All authors have read and agreed to the published version of the manuscript.

**Funding:** This research is based on work supported by COMPASS-FME, a multi-institutional project supported by the U.S. Department of Energy, Office of Science, Biological and Environmental Research as part of the Environmental System Science Program. The Lawrence Berkeley National Laboratory operates for DOE under U.S. Department of Energy Award No. DE-AC02-05CH11231. The mapping of Lake Erie coastal wetlands was funded by the U.S. Environmental Protection Agency's GLNPO office through grant (GL-00E00559-0), it has not been subjected to any EPA review and therefore does not necessarily reflect the views of the Agency, and no official endorsement should be inferred.

**Data Availability Statement:** The datasets used in this study are freely available at https://doi.org/10.15485/1876578 [29].

**Acknowledgments:** We would like to thank Dan Hawkes for his careful technical editing.

**Conflicts of Interest:** The authors declare no conflict of interest.

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
