# Peer review of "Machine-Learning Functional Zonation Approach for Characterizing Terrestrial–Aquatic Interfaces: Application to Lake Erie"

_remotesensing, doi:10.3390/rs14143285_

Round 1
Reviewer 1 Report
The manuscript presents a report on using machine learning to characterize terrestrial-aquatic interfaces. The topic is interesting and brings useful information to the readers. I have a few comments.
- The model validation part is missing, which is very important to evaluate the approach.
- All the data are from the historical database in the past, it will be great if there are some recent investigations that can fit the model.
- The conclusion part needs to stand alone.
Reviewer 2 Report
This is an excellent research investigating the ecosystem functional zone along the aquatic-terrestrial interface based on soil, vegetation, and topography. The research is of great use for coast management. I really enjoyed reading the manuscript and have only minor suggestions (see the denoted PDF).

Author Response
[Response]:
Thank you very much for your comments. Here are the answers for the specific comments denoted in the PDF (Note that the line numbers are from the original word version without track changes, where the reviewer wrote their comments):
Title. Ecosystem functional zonation and Terrestrial-Aquatic character are repetitive. Suggest change to "Machine-Learning Approach to Characterize Terrestrial-Aquatic Interfaces: Application to Lake Erie"
[Response] We agreed that the title was repetitive, we removed the word “ecosystem” but kept the “functional zonation” part that we wanted to emphasize.
l.23. wetland “land” cover
[Response] We replaced “land” with “vegetation” cover which is more specific.
l.39. highly “biologically” productive ecosystems.
[Response] We removed the word “biologically” as it was repetitive.
l.62. What does it mean?
[Response] “The highly compressed spatial scales over which TAIs processes change could explain this knowledge gap.” We agreed that the term “highly compressed spatial scales” was confusing, and changed the sentence to: “The difficulty to capture the spatial scales over which TAI gradients change could explain this knowledge gap”.
L.203. Any GIS tools for this?
[Response] We developed an algorithm for MATLAB environment. We specified the tools we used to build the grid in the revised version of the manuscript: “We develop a MATLAB algorithm to build a grid based on principles of geometry, made of lines perpendicular and parallel to the shoreline (i.e., transverse and longitudinal dimensions).”
Reviewer 3 Report
I have reviewed manuscript remotesensing-1726470 by Enguehard et al. Whenever I review a manuscript, the first question I ask is whether the manuscript reports an application of the scientific method. The scientific method involves asking questions and/or testing hypotheses. This manuscript does neither. No questions are asked, and no hypotheses are tested. Instead, the authors say (lines 107–111): we aim to study the spatial heterogeneity of coastal TAIs and plant dynamics through two objectives. Objective 1 consists of investigating the co-variability between topographic metrics, plant productivity, and soil data across the coastal wetlands in order to characterize them. Objective 2 aims to evaluate the potential of a specific unsupervised ML method, called hierarchical clustering, to establish spatial zones of similar topography and plant productivity characteristics. Studies, investigations, and evaluations are all words that might involve asking question and/or testing hypotheses, but the authors never ask any questions or pose any hypotheses. I think this manuscript would be much improved if the authors sat down and decided what questions they wanted to answer and/or what hypotheses they wanted to test and then organized the manuscript accordingly. In the context of formulating the questions and hypotheses, I think it will be important for the authors to clearly explain why it is important that we have answers to the questions they are asking and/or why it is important that the hypotheses be tested and determined to be either true or false. In its current form, I do not think this manuscript qualifies as a scientific study, i.e., as an application of the scientific method. However, with appropriate modifications, I think this manuscript would qualify as an application of the scientific method.
Round 2
Reviewer 3 Report
See attached

Author Response
Thank you for your suggestion. We have revised the manuscript according to the comments from the Editor and Reviewer 3.
We have made the hypothesis more specific to this paper and our analysis: “We hypothesize that (1) the different wetland types have distinct topographic signatures and plant productivity, and are organized as a function of the distance from the shore; and (2) we can find statistically distinct zones that capture these co-varied properties among the coastal wetlands." (Line 105-108)
In the result section, we have performed statistical hypothesis testing (ANOVA) to test whether the difference is statistically significant among the four wetland types (Line 312-314), as well as among the identified zones (Table 2, and Table 3). In these two cases, the null hypothesis is that these wetland types or the zones have the same mean values. We obtained p-values smaller than 1 x 10-10, confirming that the differences are statistically significant, and that the null hypothesis is rejected. We included these statements in Lines 312-314 and the Table 2 and 3 captions.